# Textile Knitted Stretch Sensors for Wearable Health Monitoring: Design and Performance Evaluation

**DOI:** 10.3390/bios13010034

**Published:** 2022-12-27

**Authors:** Md Abdullah al Rumon, Gozde Cay, Vignesh Ravichandran, Afnan Altekreeti, Anna Gitelson-Kahn, Nicholas Constant, Dhaval Solanki, Kunal Mankodiya

**Affiliations:** 1Department of Electrical, Computer and Biomedical Engineering, University of Rhode Island, Kingston, RI 02881, USA; 2Department of Textiles, Rhode Island School of Design, Providence, RI 02903, USA; 3EchoWear LLC, Pawtucket, RI 02860, USA

**Keywords:** knitted sensors, stretch sensors, smart textiles, e-textiles, wearable health sensors

## Abstract

The advancement of smart textiles has led to significant interest in developing wearable textile sensors (WTS) and offering new modalities to sense vital signs and activity monitoring in daily life settings. For this, textile fabrication methods such as knitting, weaving, embroidery, and braiding offer promising pathways toward unobtrusive and seamless sensing for WTS applications. Specifically, the knitted sensor has a unique intermeshing loop structure which is currently used to monitor repetitive body movements such as breathing (microscale motion) and walking (macroscale motion). However, the practical sensing application of knit structure demands a comprehensive study of knit structures as a sensor. In this work, we present a detailed performance evaluation of six knitted sensors and sensing variation caused by design, sensor size, stretching percentages % (10, 15, 20, 25), cyclic stretching (1000), and external factors such as sweat (salt-fog test). We also present regulated respiration (inhale–exhale) testing data from 15 healthy human participants; the testing protocol includes three respiration rates; slow (10 breaths/min), normal (15 breaths/min), and fast (30 breaths/min). The test carried out with statistical analysis includes the breathing time and breathing rate variability. These testing results offer an empirically derived guideline for future WTS research, present aggregated information to understand the sensor behavior when it experiences a different range of motion, and highlight the constraints of the silver-based conductive yarn when exposed to the real environment.

## 1. Introduction

Wearable electronics for smart garments have been merged into a new era through combining soft textile materials and hard conventional electronics. There is a rising interest in the materials and designs to use soft textile materials for advanced physiological and biochemical health monitoring applications [1,2,3]. One of the major challenges in smart textiles is the accuracy of sensors’ performance in terms of sensitivity, repeatability, and durability, which often compromise for external factors such as structural deformation, temperature, humidity, sweat, etc. [4,5]. At the same time, the structural variation in textiles has enabled researchers to integrate sensors seamlessly [6,7]. In recent years, professionals and hobbyists have been incorporating circuitry into textile materials using embroidery, screen printing, and inkjet printing [8,9,10]. In recent times, weaving, knitting, and braiding techniques have also gained popularity in designing in-fabric sensors [11]. Compared to the integrated rigid sensor [12,13,14,15], the in-fabric sensor design refers to a sophisticated technique where the sensing materials are seamlessly incorporated into the textile fabric during the fabric manufacturing process [16,17,18]. One particular in-fabric sensor that has attracted attention is the stretch or strain sensor. It has been used to monitor different types of physiological health, and body movement applications [19,20,21]. Generally, strain sensors have been designed to convert a mechanical input into a measurable electrical signal. The change in electrical values can be capacitance [22], resistance [23], or inductance [24]. Most textile-based sensors work in a change of resistance. Thus, the mechanism relies on a resistive strain gauge [25,26]. However, there are other textile-based sensors that work in a change of capacitance [27,28]. Textile-based strain sensors can be developed at multiple structural levels, such as yarn [29,30,31], fabric [32,33] or even in spacer [34] fabric form. Knit fabrics are the most suitable structures to integrate strain sensors due to their inherent stretchable properties. Prior research showed the demand for using knitted stretch sensors in versatile applications, for instance, communications [35,36], biomedical applications [37,38,39], energy storage [40], etc. The adoption of knitted sensors is slow and facing several challenges. The shortcomings being explored include their non-linearity and hysteresis in response to applied strain, as well as the drift of the electrical characteristics of the textile sensor with time and use.

For medical applications, the range of stretching can be divided into two types: micro and macro. The first type refers to those applications where finer and small stretching is required, for instance, respiration [41,42,43]. The second type relies on extended stretch-ability, for instance, body motion [44] and knee monitoring [45,46]. All the research showed the potential possibilities of knitted sensors in different applications, but real-life exposure may limit the sensing capabilities of those sensors. Textile structures are soft and stretchable and can be easily compromised by the external environment, such as water and sweat. In addition, the wear-ability and human ergonomic factors always create challenges for the experimental stretching and real-life stretching of the sensor. Designing and developing sensors after understanding the core behavior of the structures and constraints of the materials will create a solid base for real-life applications.

In this work, we focused on investigating how the sensitivities of six plain-knit structures differ from one another due to regular and repeated mechanical stretching and releasing actions. We also explored the effect of external environmental stimuli such as water and sweat. The research tried to illustrate variable changes the sensors’ linearity during stretching. The characterization includes the sensors’ electrical, mechanical, and chemical performance in simulated testing. We developed a custom motorized gear and rack setup to quantify the sensor’s different stretching with a precision of 5 mm. We tested the sensors with salt water and analyzed the effects using SEM and EDS elemental analysis. The results show the chemical stability of the silver thread that we used for sensor fabrication. In addition, we performed 1000 cycles of stretch testing to understand how and why the sensing range drifts over long-term use.

We also developed a knitted belt system and performed a breathing test with 15 healthy participants. The test consists of three respiration (slow, normal, rapid) protocols, where we analyzed time variability, error percentage, and correlations among those breathing types. Moreover, we investigated the sensor performance of sensing inhale and exhale cycles for different participants.

The major contributions of this research are:A comprehensive demonstration of knitted stretch sensors with different structures and their performance.Investigating a setup showing different stretching percentages of knitted sensors and identifying how knitted loop structure changes the sensor’s sensitivity.Study and analysis of the sensor’s durability and chemical stability to quantify sensing materials constraints.Demonstration and design recommendations of the knitted sensors in respiration monitoring applications.

## 2. Materials and Methods

### 2.1. Machine

Knitting is a type of fabric manufacturing process composed of interconnected loops. This loop formation process occurs in only one direction, which might be horizontally (in weft knitting) and vertically (in warp knitting). Circular and Flatbed are two major kinds used mostly in the industry to manufacture weft-knitted fabric. The flatbed machine has a carriage that can move forward and backward, known as the Head or Cambox works to construct the knit, tuck, and transfer stitches. This type of machine can make complex and sophisticated knit designs. We used a Stoll-Flatbed knitting machine to develop the sensors (gauge-10) [47]. The machine can feed multiple yarns in each bed using double-hooked latch needles. With the use of a set of sliders in each bed, the single pair of needles can be moved to knit in one of two needle beds that are directly opposite one another. The machine has advanced M1 Plus [48] Knitting Software that ensures production benefits by generating patterns for a highly optimized knitting process. Three plain structures (also known as single knit) were used for the sensor: 1 × 1 stripe, 1 × 2 stripe, and solid (Figure 1). The structure is produced by only one set of needles with all the loops intermeshed in the same direction. Figure 1 shows the notation (single repeating unit) of the knit structure, which was given as the input instruction to the M1 plus software. Figure 2 represents the overview of the structural design, where the wale is a column of loops running lengthwise, and the course is a crosswise row of loops.

### 2.2. Materials

For the fabrication, we used silver-coated polyamide yarn (Shieldex^®^ 117/17 HCB) sourced from Shieldex [49] (resistance < 500 Ω/m) and polyester yarn (100%) sourced from Madeira USA [50]. This specific conductive yarn was chosen because it has a longer staple fiber which created a better uniformity compared to other available stainless steel-based conductive yarn. The other reasons are friction and yarn breakage percentage. The stainless steel yarn consists of short-stranded fiber, which creates excess projectile fiber (extra fiber around the yarn) during spinning, which results in extra friction during fabrication and generates heat and yarn piling. The silver-coated yarn consists of multiple polyamide filaments with a titer of 117dtex, which are evenly coated in silver nanoparticles and spun together.

### 2.3. Design

Three plain knitting structures were used to develop six sample sensors. Plain knitting can be recognized by its flat, uniform appearance having interlocking ‘v’ shapes on the front and crescent shapes on the back. Despite the endless design possibilities with the Stoll knitting machine, these three types were tested as the first step towards understanding the functional behavior of knitted sensors to provide a solid foundation from which deviations and explorations can continue in the future. These three structures were selected based on understanding the effect of positioning conductive yarn (CY) and polyester yarn (PY) during stretching. In addition to developing one 1 × 1 and 1 × 2 stripe sensor, we made three solid structures (different dimensions) and one hybrid solid structure (Table 1). Figure 3 shows all the sensors’ structures.

For both 1 × 1 and 1 × 2, the stripe structures (sensor 1, 5) show an alternate set of PY and CY in course direction (Figure 3). In the 1 × 1 design, the CY set was fed exactly after the one set of PY. Similarly, in the 1 × 2 structure, the machine fed CY after two sets of PY. The stripe design increases the resistance of the sensors, as the PY directly interferes with the connection between successive conductive loops. However, the stability of the structure increases as the PY is coarser than the CY. The solid design consists of only CY throughout the successive loops (sensor-2,3,4). This close connection between conductive loops drops the resistance. Three different dimensions of the solid design exhibit a comparative understanding of how sensing capabilities vary within the sizes.

Sensor-6 shows a hybrid yarn combination of CY and PY. This structure is a combination of the other two types and showed a relatively high stability and less resistance (result section). The CY and PY were fed simultaneously during the knitting process. We used different colored PY for the sensor to understand the visual difference.

### 2.4. Sensing Mechanism

The knit structure is composed of successive intersecting loops, where all the loops are formed by a single set of yarn in the course direction. A single set refers to one or multiple yarn feeds. Woven fabrics, on the other hand, are made by interlacing yarns (warp and weft) in a perpendicular direction. Where each of the warp yarn (lengthwise yarn) connects to the loom separately, this single-set continuous loop formation allows adding the conductive yarn in the middle of the fabric more easily than the woven fabric.

Each knit loop can be divided into three parts: head or top-arc (top curve), legs or side limbs (middle part), and feet or the bottom half-arc (bottom curve) (Figure 4). Two sets of longitudinal loops (wales) connect by the feet of the first loop and the head of the second loop.

Stretchability is a vital characteristic of the knit structure. The higher elastic deformation capabilities allow this structure to use for those applications where we need repetitive stretching. Stretchability mostly depends on the loop structure, size, and the kind of yarn that is used. Stretching on either side (course or wales) of the fabric creates more interaction between subsequent arcs; however, course-wise direction usually shows more stretchability than the wales, as the curved arcs can facilitate extra length when stretched.

In our testing sensor, we used silver-based conductive yarn. Figure 5 shows that the sensor fabric is stretched in the course direction, and a structural change happens on the two conductive successive loops and creates more contact points between the top and bottom arc; thus, a shorter path is created for the current to move from power to the ground, and therefore shows a lower resistance value than when the sensor is at rest. The amount of contact points is proportionally related to the stress of longitudinal stretching.

## 3. System and Experimental Design

### 3.1. Electromechanical Test Setup

An electromechanical stretching system was developed to simulate repeated stretches. The motorized system is specifically designed to hold the stretch sensors and provide a repetitive loading and unloading cycle. A universal testing machine (UTM) was an option, but there were two constraints: the first one was UTM machines, which usually come with a vertical setup where the specimen experiences the gravitational force erectly; the second one was, in our testing, we kept the specimen horizontally, and during the salt fog test, we applied the salt solution, which might be a problem for the sophisticated UTM machine. The system included (1) two Arduino microcontrollers; (2) an external 9 V DC power source; (3) an L7805 voltage regulator; and (4) a Hitec HS-788HB [51] servo motor with a stall torque of 13.2 kg-cm (Figure 6). The system (rack-gear) mechanism has a minimum longitudinal resolution of 5 mm. The knitted sensor was clipped on one static side and the other movable (rack and gear) side. We set up a unique set of instructions for each test with the servo motors.

The specific stretching length for each stretch percentage was calculated using the following Equation (Equation 1), where *δl* is the stretching length, *l* is the initial length, and *n* is the stretching percentage of the sensor. The value was used to program the gear setup for precise stretching.
(1)$$\delta l=\frac{l\times n}{100}$$

### 3.2. Stretching Test

Prior studies showed the possibility of using the knitted sensor for diversified physical activity monitoring applications [45,52,53,54]. Each application showed different sensing capabilities for micro stretch monitoring, such as respiration, and macro stretch monitoring, such as body joints (elbow and knee). We designed a test with different stretching percentages (10, 15, 20, 25) having 5 mm of stretching resolution in our experiment. Multiple loading showed the relation between the stretch amount, mechanical deformation, and sensitivity. To understand the core changes of the sensors, we performed two tests—intermittent and continuous stretching.

For the intermittent test, we initiated by relaxing each sensor in its sedentary position and recorded the resistance. Then, we stretched the sensors individually in different stretches and held each position for 60 s.

We separately calculated the mean resistance of the sensor using Equation (Equation 2), where R(o) is the mean resistance of the sensor during sedentary position, *n* is the number of samples, and *i* is the sample resistance. Equation (Equation 3), R(p) refers the mean resistance during different stretch points *p* (10, 15, 20, 25)%.
(2)$$R\left(o\right)=\frac{\sum_{i=1}^{n}X{\left(o\right)}_{i}}{n}$$
(3)$$R\left(p\right)=\frac{\sum_{i=1}^{n}Y{\left(p\right)}_{i}}{n}$$
(4)$$\Delta R=\frac{\sum_{i=1}^{n}X{\left(o\right)}_{i}-\sum_{i=1}^{n}Y{\left(p\right)}_{i}}{n}$$

Then, we measured the resistance difference (RD) using Equation (Equation 4), where ΔR was calculated by subtracting Equations (Equation 2) and (Equation 3). The resistance difference is an arithmetic expression that depicts the overall resistance changes between two specific stretch positions. If the sensor shows a gradual and consistent increase in RD, it means that the sensor has good linearity throughout the stretching. Any sudden drop in the RD due to stretching shows the constraints of the sensor’s linearity and sensitivity.

For the continuous stretching test, we performed different stretches %(10, 15, 20, 25) 10 times. The test data show a continuous periodic change in resistance value due to loading and unloading.

### 3.3. Durability Test (Cyclic Test)

The knitted structure is made up of intermeshing loops, which usually try to return to their original shape when the stress is released. However, the phenomenon is subjected to structural deformation when the stress continuously happens (either side of the course or wale direction) over a long time. This cyclic loading and unloading create plastic deformation on the material’s layer and propagates throughout the whole structure.

This scenario impacts applications that require repetitive motion such as respiration or body movement. Thus, we aimed to investigate the effects of repetitive movements on sensor performance. Each sensor was continuously stressed and released from 0 to 25% of its length in this experiment for 1000 cycles. We evaluated the sensing range and how the baseline shifted over time and during the cycle.

We measured the gauge factor (GF) of all sensors before and after the durability test. GF, also named the sensitivity coefficient, is usually used to evaluate the sensitivity of strain sensors. It is the ratio of the fractional change in electrical resistance to the fractional change in length (strain) which can be computed by the following Equations (Equation 5) and (Equation 6).
(5)$$\varepsilon =\frac{l_{T}-l_{0}}{l_{0}}$$
(6)$$GF=\frac{\Delta R/R_{0}}{\varepsilon }$$

In Equation (Equation 5), ε, $I_{0}$, and $I_{T}$ represents the strain, unstretched sensor’s length, and stretched sensor’s length, respectively. In Equation (Equation 6), $R_{0}$, and ΔR represent the initial resistance (before stretch) and test resistance (after stretch). The sensitivity coefficient GF was calculated from the relative resistance variation of the sensor versus its elongation (25%) at moment T.

### 3.4. Salt Fog Test

In real life, sweating is a common issue that leads to chemical changes in garments. Usually, eccrine glands produce most of the sweat and consist primarily of water (99%, and a small amount of sodium, potassium, chloride, and other essential minerals (1%) [55]. For smart textile applications, it is predictable that the in-fabric sensors will go through some form of electrochemical corrosion when the current passes through the floating ions in the sweat. In this test, we conducted two experiments: the first one was conducted by applying water on the sensor, and in the second test, we applied salt water solution on the same sensor. Before the water test, we first stretched and released (0–25%) the sensor 10 times and measured the continuous resistance. Then, we sprayed water, waited 10 min, repeated the same testing protocol, and measured resistance. For the salt-fog test. We made a solution with 5% sodium chloride (NaCl) and sprayed it on the sensor using a pressurized sprayer, which gave a uniform diffusion of salt water throughout the sensor. We kept it in the lab environment for 60 min to settle down the sample. In the same way, we stretched and released (0–25%) the sensor 10 times and measured the continuous resistance.

The salt solution was prepared by dissolving 25 g of dry NaCl sourced from Sigma-Aldrich [56] into distilled water and making a 500 mL solution. The pH of a NaCl solution remains 7 due to the extremely weak basicity of the Cl- ion, which is the conjugated base of the strong acid HCl.

After testing, we kept the sample for 24 h, allowing sufficient time for any electrochemical reaction to happen, and did an imaging test to understand the changes and morphology of the silver-based conductive fiber.

## 4. Results and Discussions

### 4.1. Investigating Effects of Stretching on the Sensor Performance

The stretching test results are divided into two segments. One is intermittent stretching, where the mean resistance difference was calculated when the sensor was stretched and held in position for 60 s. Furthermore, the second is the continuous stretching and releasing of the sensor without any rest. Both of the tests were carried out with 10 cycles.

#### 4.1.1. Intermittent Stretching

For this timed stretching test, we measured two mean resistances of the sensor *R(o)* (Equation (Equation 2)) and *R(p)* (Equation (Equation 3)) when it was unstretched and stretched, respectively. Table 2 shows the mean resistance differences of each sensor generated from Equation (Equation 4). According to our previous structural explanation, we predicted that sensor-1 and sensor-5 (1 × 1 and 1 × 2 stripes, respectively) should exhibit more resistance because those structures fed conductive yarn (CY) alternately after one and two polyester yarn (PY) yarns for sensor-1 and sensor-5, respectively. Similarly, all the solid structures (sensor-2, sensor-3, and sensor-4) should show less resistance because of their firm CY loops. We were optimistic about the sensor-6 result that should be between the stripe and solid as we blended the CY and PY yarn together during feeding. As we expected, the testing results showed similar changes until all the sensors were stretched until they exceeded 15%. After that, stripe sensors behaved randomly. According to our hypothesis, this happened because of the internal dissemination of charge between the conductive yarn (made with polyamide) and polyester yarn. Naturally, polyamide and polyester material shows positive and negative charges (respectively) when they experience friction [57,58]. When the striped sensors were stretched more, there was more friction between the CY and PY, which resulted in activating the charge dissemination between the yarn. This dissemination process acts in different stretching percentages for 1 × 1 and 1 × 2 stripe sensors.

Figure 7 shows that for the solid sensor-3 and sensor-4, their changes were linear but worked poorly until 15%; after that, both showed good resistance changes. This can be attributed to their close and successive conductive loop connection. On the other hand, despite being a solid sensor, sensor-2 worked well. Logically, this happened because of the size of the sensor, which allowed a better resistance drop when it was stretched. There is a step-up behavior resistance drop and the sensor size of the solid structure. The sensitivity increases in function of the size of the solid structure. Sensor-6 showed the most linear resistance changes through the different stretching (Figure 7 The blended yarn combination gave the sensor extra structural stability, and the mixed yarn (CY and PY) combination made the charge dissemination process neutral.

#### 4.1.2. Continuous Stretching

The continuous cyclic stretching test is the extended version of the previous experiment. All the graph depicts similar characteristics as we found earlier (Figure 8). One of the significant findings is how the sensor’s baseline shifts over and over for different stretches. Baseline changes refer to the shifting of the resistance value from its initial position with the same stretching percentages. This is a common structural limitation of textile structures. Knitting is a fabric manufacturing process of intermeshing loops. When the knit structure is stretched, the inside changes are mechanical, where the distance between the top arc and bottom arc changes laterally and longitudinally. Usually, it takes longer return it to its original position. However, the phenomenon and the time duration depend on the materials used for knitting. Here, we used silver-based yarn for the sensor. The yarn surface is rough and twisted with fibers, which creates friction and extends the time to come back after stretching.

Among all the sensors, sensor-6 showed the most continuous and stable resistance increment and minimum baseline shifting. Sensor-1 and sensor-5, both stripe sensors, showed good sensitivity until 15% and 20%, respectively. Other solid sensors behaved with irregular sensitivity. In the prior test result, we saw that sensor-2 worked well among all the solid structures; here, it did the same, although the baseline shifted drastically.

### 4.2. Durability Test (Cyclic Test)

We assume that the sensor is going through repetitive movement, where stability is the key factor to provide reproducible sensing performance. All the sensors were repeatedly stretched and released for this test. The previous experiment showed how the cyclic (10) test changes the sensor’s baseline. This durability test revealed some crucial insights into the sensor behavior and showed a pathway for possible applications. Figure 9 shows the accumulative stretching error (baseline) that happened on each of the sensors after cyclic stretching 1000 times from 0 to 25%. For Figure 9, we measured the unstretched and stretched mean resistance at cycle-1 and cycle-1000 using Equations (Equation 2) and (Equation 3), respectively, and plotted all the results together. The yellow and blue line indicates how the base value changed over cycles of continuous stretching and unstretching, respectively. Furthermore, the corresponding green and red dots represent the cyclic status (green means the unstretched resistance value during the first cycle, and red means the value after the 1000 cycle). During the cyclic test, all the sensors showed a linear shift in their baseline (stretched and unstretched). This change represents the structural deformation of the sensor, and a slow restoration time, which means that the sensor did not have enough time to get back to its original shape. Sensor-6 showed the maximum linear changes of its stretched and relaxed position, which also represents the higher sensitivity of the sensor (Figure 10).

Table 3 shows the sensing coefficient gauge factor (GF) of all sensors. All the results were derived using Equations (Equation 5) and (Equation 6). From a theoretical point of view, the gauge factor is defined by the function of its mechanical and electromechanical properties. All the sensor readings demonstrated a nearly identical GF. Usually, the GF range of most knitted strain sensors is between 0.42 and 5 [59]. A high gauge factor indicates a large change in resistance, which leads to high sensitivity.

### 4.3. Salt-Fog Test

The test result was calculated by obtaining the mean resistance of the sensor when it was stretched (25%) and released in the air using Equation (Equation 4). In the same way, we measured the resistance after applying the salt solution to the sensor.

Figure 11 shows the salt-fog test (SFT) error chart. Where all the sensors showed a considerable amount of base resistance drop (unstretched state). The presence of NaCl(l) in the solution becomes Na+ and Cl− ions and assists in passing electrons, which results in a resistance drop.

In this chart, the orange line indicates the changes in the resistance of the sensor when it was not stretched (base value). The blue and red dots represent the average sensor resistance (10 cycles) in the air and the salt water, respectively. Similarly, the green line depicts the resistance changes while the sensor was stretched 25%. The chart shows that sensor-6 made a substantial drop in resistance.

Figure 12 shows the detailed changes of sensor-6 when it was introduced with water and salt separately. The gradual drops in resistance indicate that the silver-based knitted sensor has an inverse correlation with its sensitivity and the number of floating ions. Other sensors’ (Sensor 1–5) results are added in Section A.2.

In other tests, we sprayed over the sensor and kept it for 60 min without water-wash; the microscopic view shown in Figure 13, where the silver on the yarn was washed away due to chemical corrosion, occurred due to the Cl- icon reacting with Ag+ (conductive thread) and precipitating silver chloride (AgCl), which destroyed the sensor’s conductivity.

We also observed the morphological changes in the silver yarn that happened after SFT. Figure 14 shows scanning electron microscopy (SEM) images, where Figure 14a represents the yarn before SFT. The brighter outside layer represents the silver coating. On the other side, Figure 14c shows a clear change that happened after SFT. The silver layer was washed out, and the dark layer polyamide (core fiber) layer came out. To analyze the distribution of the material changes that happened before and after SFT, the elemental energy-dispersive X-ray spectroscopy (EDS) analysis was conducted. Figure 14b shows a higher weight percentage (27.5) of silver (Ag), and Figure 14d shows a minute weight percentage (1.4) of the presence of silver (Ag). The other elements of this conductive yarn are C, N, and O. The change in silver material depicts a clear conclusion of the instability of the silver-based conductive yarn during the interaction with salt water.

### 4.4. Hysteresis

Hysteresis shows dynamic changes in a system of how the output correlates with the input values. For the knitted sensor, the hysteresis graph shows whether the sensor has good stability and repeatability. However, the result is dependent on other variables including the number of cycles, stretching percentages, and external stimuli (water, salt). In our test, we used a sample with repetitive 10-cyclic loading and unloading applied to different stretching %(10, 15, 20, 25). The average changes were plotted in the graph and shown as a routing for loading and unloading.

The previous continuous stretching result (Figure 8) showed the changes and repeatability of all the sensors, where sensor-6 showed the most stable changes due to its structural stability. Here, Figure 15 shows the hysteresis result of sensor-6. The result exhibits a minute amount of hysteresis over different stretching amounts. The result was very satisfying and concluded the capabilities to regain its original shape and structure. Other results are added in the Section A.3 section.

## 5. Sensor Testing—Regulated Respiration Monitoring

After all the evaluations of the sensors, we found a clear difference between sensor-6 and the rest of the sensor types in terms of accuracy, repetitiveness, and sensing range (macro–micro). For that, we used sensor-6 for further human testing. We tested 15 healthy participants (11 males and 4 females) with an age range of (24–32) and we tested them with three breathing rates (BRs)—slow, normal, and fast [IRB2122-009]. Slow breathing, known as bradypnea, refers to a breathing rate of less than 12/min for adults [60]. Normal breathing refers to a relaxed state, with a breathing rate of 12–24/min [60]. Fast breathing, known as hyperventilation or over-breathing, refers to a breathing rate of more than 24/min [60].

### 5.1. Testing Protocol

The testing protocol was designed with the requirement to understand sensing capabilities. We asked participants to stand up and follow three visual lap timers consecutively representing different types of breathing (inhale–exhale); 1: slow (6 s/breath), 2: normal (4 s/breath), and 3: fast (2 s/breath). Each breathing test consists of 10 cycles, which means that the required time for slow breathing is 60 s (10 breaths/min), normal breathing 40 s (15 breaths/min), and fast breathing 20 s (30 breaths/min) [60].

### 5.2. Data Acquisition System

We used sensor-6 and made a belt using a stretchable ribbon. Figure 16 shows the data acquisition system, where we used two ESP32-S2 Feather boards sourced from Adafruit, USA [61]. One board was connected to the belt, and the other one was connected to the computer. We used the ESP-NOW Wifi communication system for hassle-free wireless data acquisition. The MCU unit was attached to the belt using velcro and connected to the sensor using a snap button. The sampling rate was set to 75 Hz.

To measure the breathing rate, we used the following Equation (Equation 7). The *n* refers to the number of the breathing cycle (inhale-exhale) and B(t) represents the total time taken for a *n* number of breaths. For this test, *n* is equal to 10.
(7)$$BreathingRate=\frac{60\times n}{\sum_{i=0}^{n}B{\left(t\right)}_{i}}$$

### 5.3. Test Result

All the participants’ breathing raw data were processed using Matlab’s low-pass Gaussian filter library (smoothing factor 0.2) to reduce noise (high-frequency components) in the signal. We also used max and min indices functions (prominence-9) to point out the peaks for inhale and exhale. Figure 17 shows an example data (participant-9). The graph shows a clean peak detection for both inhale and exhale. All the other participants’ data were added in Section A.4.

Figure 18 shows the time variability results for 15 participants. Each of the dots represents an average time (10 cycles) taken for the corresponding breathing type (slow/normal/fast). The graph represents the fact that all the participants had some range of time variability compared to the referenced breathing rate in spite of following the breathing lap timer. However, the result shows a distinctive time variability between breathing types (slow—60 s, normal—40 s, fast—20 s), which concludes a satisfactory linear fit and R-squared value of 0.87918.

Secondly, Figure 19 illustrates the variability of BR. We calculated the BR using previous testing data and Equation (Equation 5). In this graph, each of the dots was plotted as an average breathing rate of 10 cycles. The error bar (5%) represents the comparative uncertainty or variation of the corresponding referenced value. Figure 19 displays an error chart of the breathing rate variability for all breathing types (slow, normal, fast). All participants tried to maintain the referenced BR for slow and normal. Conversely, during fast breathing, we noticed that none of the participants could follow the referenced breathing rate of 30/min. This is reasonable given that, under normal circumstances, humans do not breathe so quickly until the brain stimulates the muscle system.

Table 4 shows comparative R-squared values for six breathing incidents. All the values represent a considerable relation to satisfy the predictable variation of the breathing rate.

The preliminary data of respiration monitoring is a promising start. More investigation is required to test the sensor’s performance on more human participants with various body sizes and ages.

## 6. Conclusions

This paper presented a performance evaluation of six knitted sensors followed by three different plain knitting structures. Polyester and silver-based conductive yarn were used for constructing the sensors. Each sensor was tested with intermittent and continuous stretching and unstretching systems covering 10–25% of the sensor’s length as a minimum of 5 mm. The test results showed the comparative performance analysis between 1 × 1 stripe, 1 × 2 stripe, solid, and hybrid structure. The durability test was performed by doing a similar stretching and releasing maneuver 1000 times. Given the importance of external stimuli, we performed the water and salt-fog test.

We discussed the comparative linearity and repeatability among those sensors. Sensor-6 was made out of a hybrid structure of polyester and silver-based conductive yarn and performed better due to its structural stability and neutral charge status. The durability test error chart showed cumulative results of resistance variation after 1000 cycles. All the sensors showed a clear jump in their resistance during this test. The phenomena represent the change in knitting structures during the test. The continuous and intermittent cyclic loading and unloading test showed how time affects the variation. More time between stretching and unstretching means that the sensor will have sufficient time to regain its structures. However, sensor-6 showed the maximum variations in this test. We also demonstrated gauge factors for all the sensors (before and after the durability test).

The salt-fog test characterizes the possible interaction of the sensor with sweat and water. The aggregated result showed the impacts on all the sensors due to the electrochemical changes on the sensor. Initially, all the sensors dropped their resistance, but after 24 h, the resistance became high due to the precipitation of silver (Ag) while reacting with between Cl− ions. The SEM and EDS elemental analysis showed the visual comparison and change in the silver yarn before and after the salt-fog test.

There are several ways of solving this problem. In this research, we constrained our focus to silver-based conductive yarn. Comparisons among different conductive yarns (carbon, copper, stainless steel) during durability and salt-fog tests will reveal more insights to uncover the material selection process for the sensors. Furthermore, there are other structures (for instance, rib, purl, and interlock) that can be evaluated. Each knit structure and material has different stretchability and stability and will show the benefits and limitations for further research. However, on the side, coating and gluing are common ways of insulating knitted structures, but in that case, the challenge is that the knitted structure may lose its natural stretchability (which limits the stretching or changes the hysteresis). A hypothetical solutions may be adding a soft silicon layer to the structure. This would serve two purposes, firstly the layer working as an insulator, and secondly, as it will work as a secondary pull-up spring to assist the whole structure in returning to its original shape as soon as the stretch is released.

The most linear sensor (sensor-6) was used to create a sensor belt for healthy human respiration testing. The test includes a guided protocol for slow (10 breaths/min), normal (15 breaths/min) and fast breathing (30 breaths/min). Fifteen healthy adult humans participated in the breathing test. The breathing time variability showed a linear fit with an R-squared value of 0.87, which represents good correlation in identifying the specific breathing types. The individual breathing rate error chart showed considerable variation in the breathing rate of different participants. The further comparative R-squared analysis chart indicates a good correlation between the breathing types. However, human testing did not have a versatile body shape and age variations study. The future study will cover greater variability in the knit sensor for an optimized performance evaluation.

## Figures and Tables

**Figure 1 biosensors-13-00034-f001:**
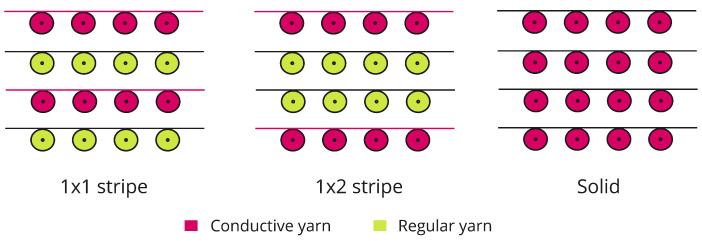
Technical notations for the knitted structures: 1 × 1-one conductive and one non-conductive yarn, 1 × 2-one conductive and two non-conductive yarns. All the yarn is conductive and solid.

**Figure 2 biosensors-13-00034-f002:**
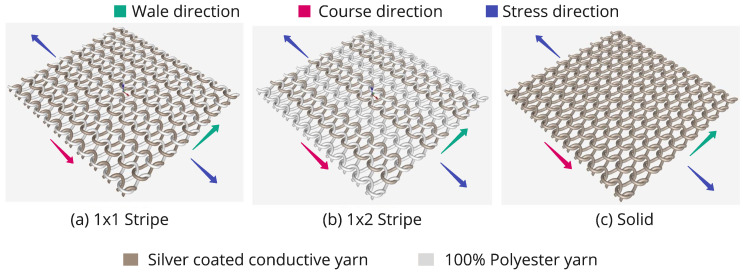
Simulation of plain knit structure composed of conductive yarn and 100% polyester yarn (non-conductive).

**Figure 3 biosensors-13-00034-f003:**
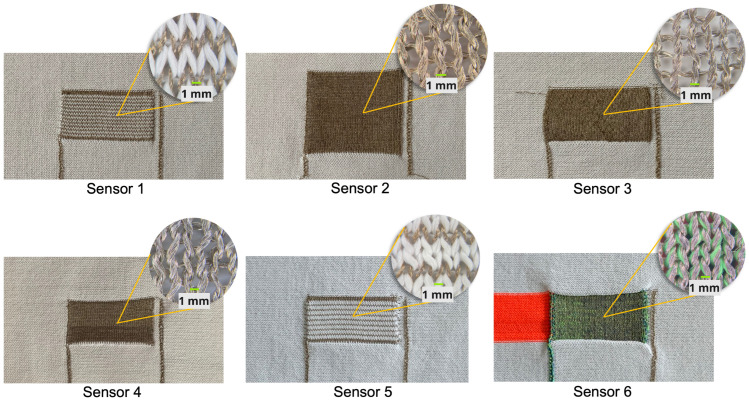
Knitted sensors. Sensor-1 (1 × 1), Sensor-2 (solid), Sensor-3 (solid), Sensor-4 (solid), Sensor-5 (1 × 2), and Sensor-6 (hybrid solid).

**Figure 4 biosensors-13-00034-f004:**
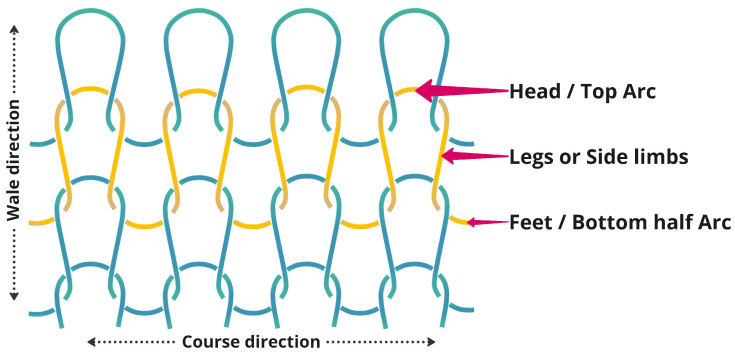
Plain knit structures and components.

**Figure 5 biosensors-13-00034-f005:**
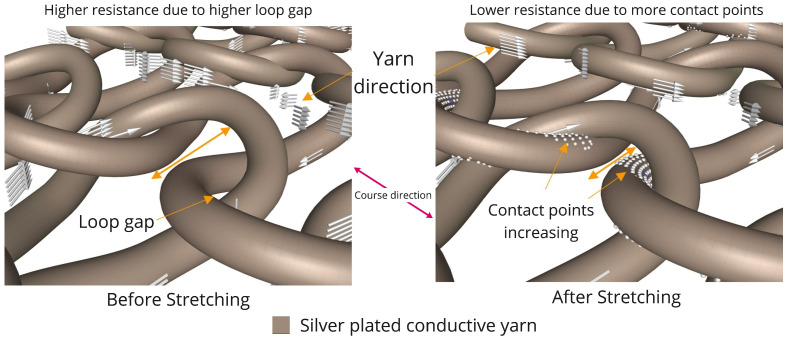
Simulation of the knitted sensor’s sensing mechanism. Contact points between the top arc and bottom arc before stretching and after stretching.

**Figure 6 biosensors-13-00034-f006:**
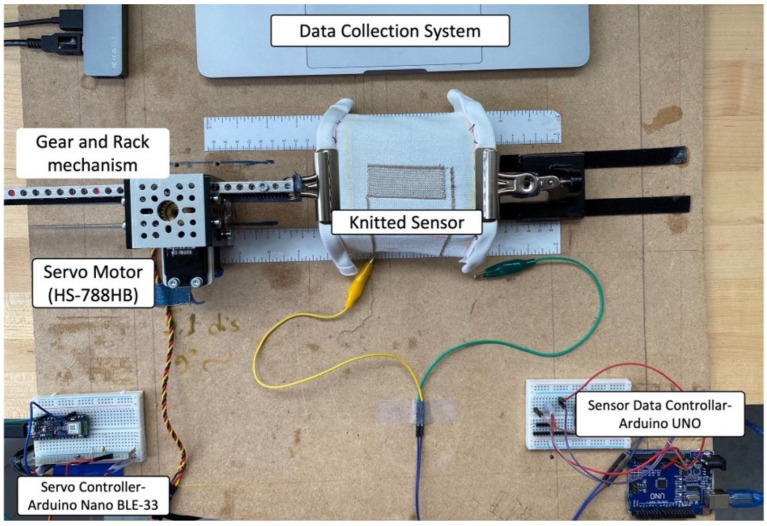
The in-lab designed electromechanical setup for cyclic tests.

**Figure 7 biosensors-13-00034-f007:**
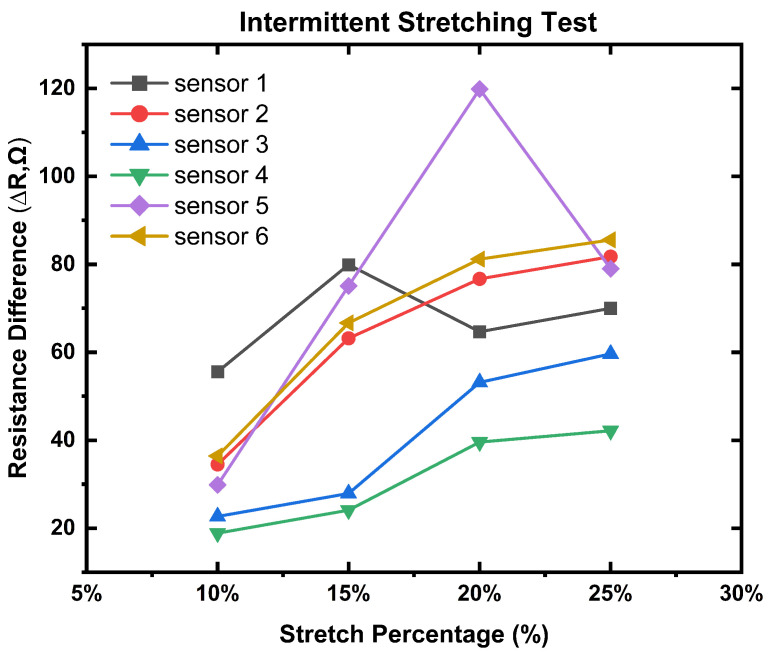
Resistance difference of all the sensors after different stretching.

**Figure 8 biosensors-13-00034-f008:**
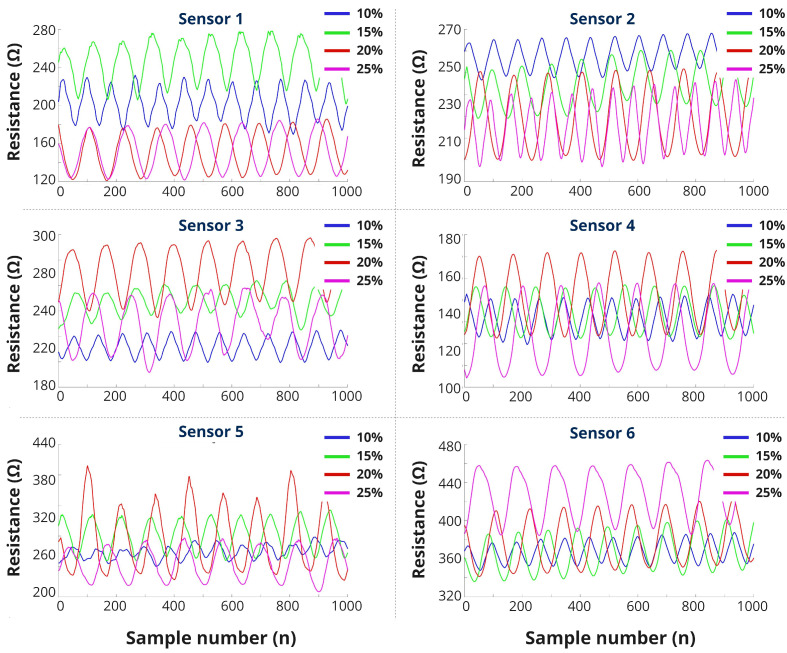
Continuous stretching (0–25%) test results.

**Figure 9 biosensors-13-00034-f009:**
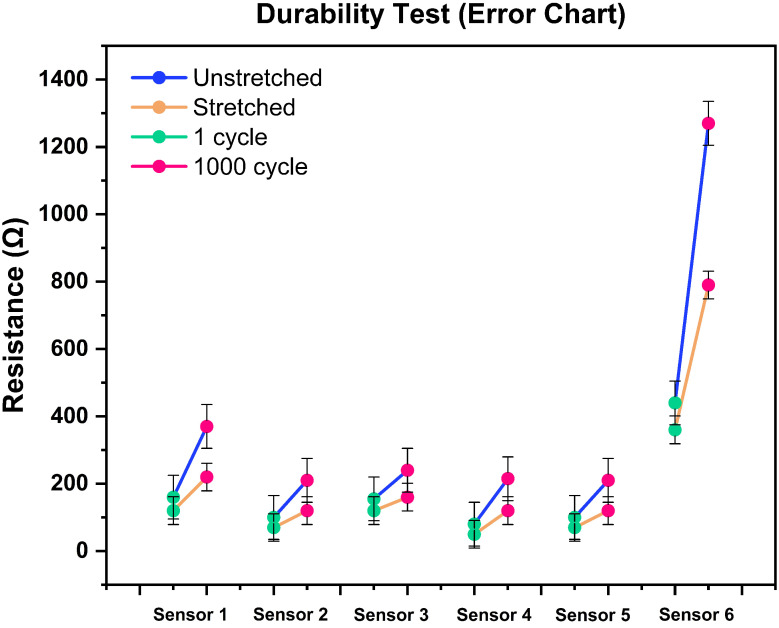
Durability-test error chart. Changes in resistance (before and after stretching 25%) in 1 cycle and 1000 cycles.

**Figure 10 biosensors-13-00034-f010:**
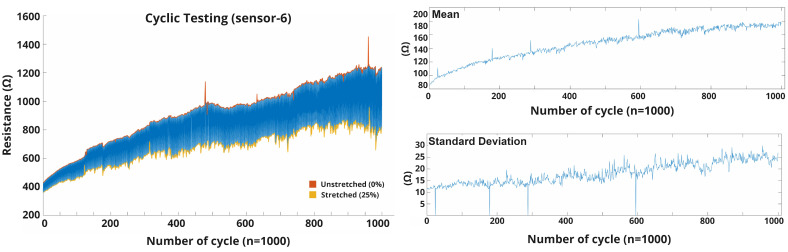
Durability test of sensor-6 (1000 cycle).

**Figure 11 biosensors-13-00034-f011:**
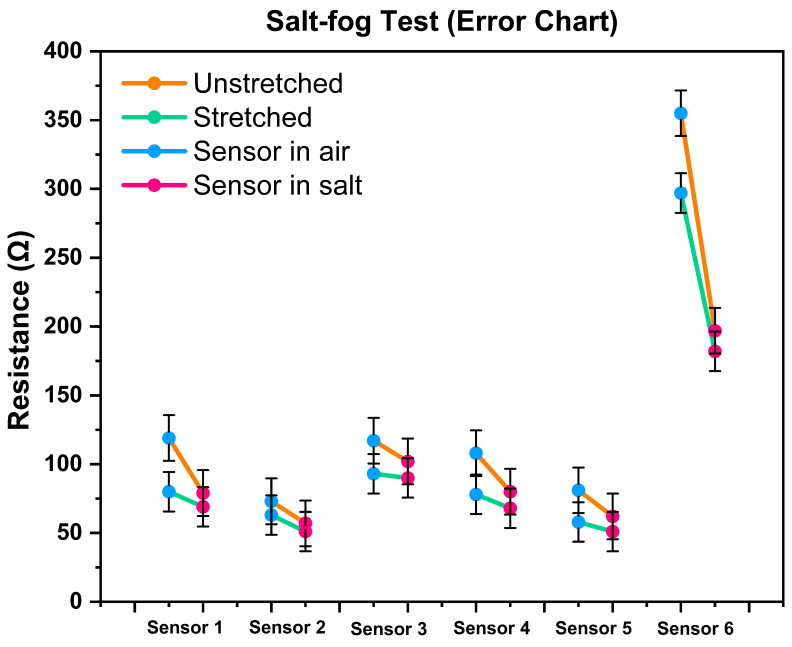
Salt-fog test error chart. Change in resistance before and after interaction with salt water.

**Figure 12 biosensors-13-00034-f012:**
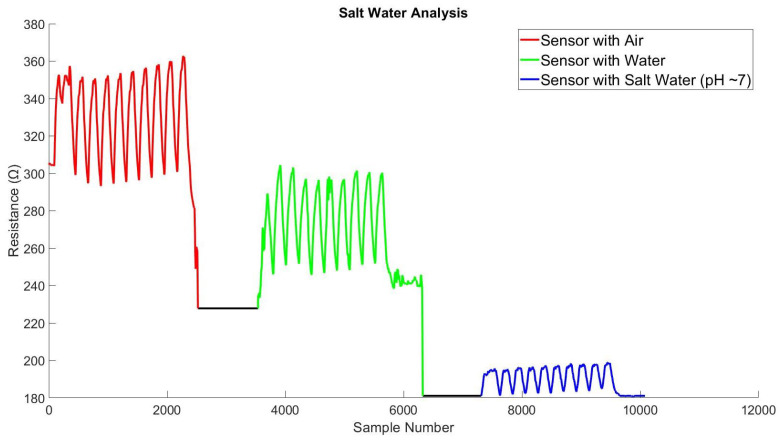
Resistance changes (sensor-6) during interaction with salt and water.

**Figure 13 biosensors-13-00034-f013:**
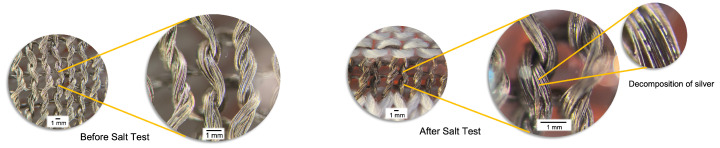
Electrochemical changes of silver-based conductive yarn after the interaction with salt water.

**Figure 14 biosensors-13-00034-f014:**
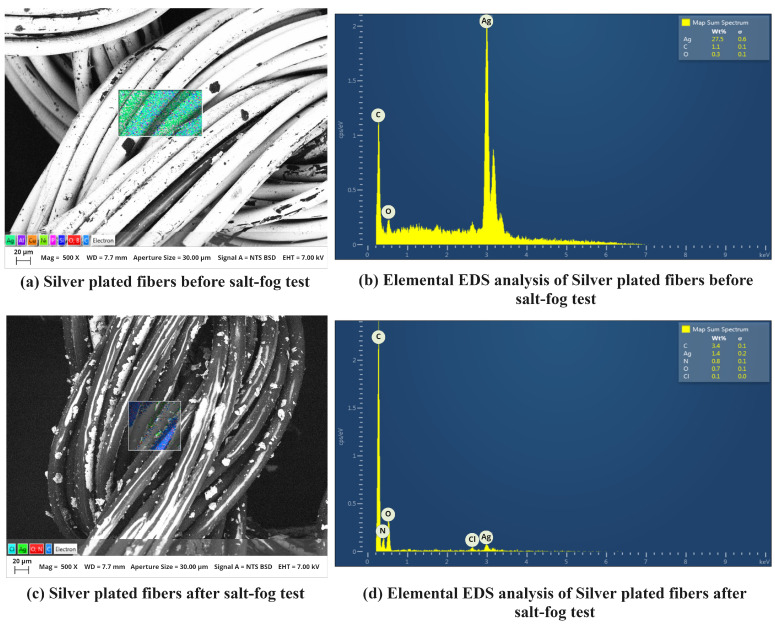
SEM images and elemental EDS analysis of the silver-coated fiber before salt-fog test (**a**,**b**) and after salt fog test (**c**,**d**).

**Figure 15 biosensors-13-00034-f015:**
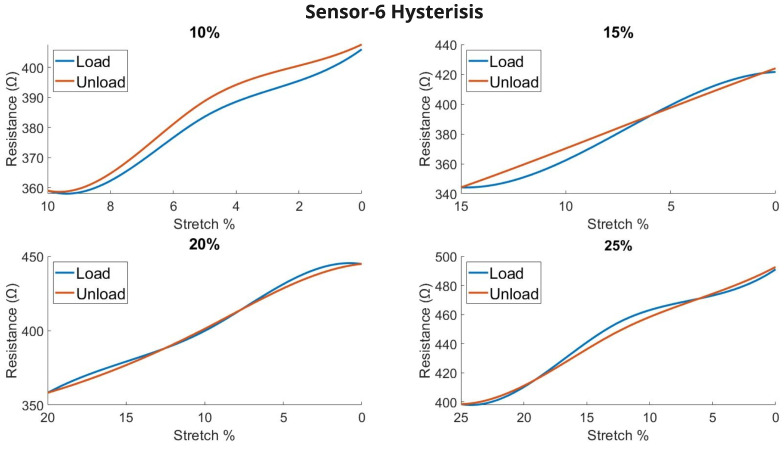
Hysteresis of the knitted sensor (sensor-6) for different stretching %.

**Figure 16 biosensors-13-00034-f016:**
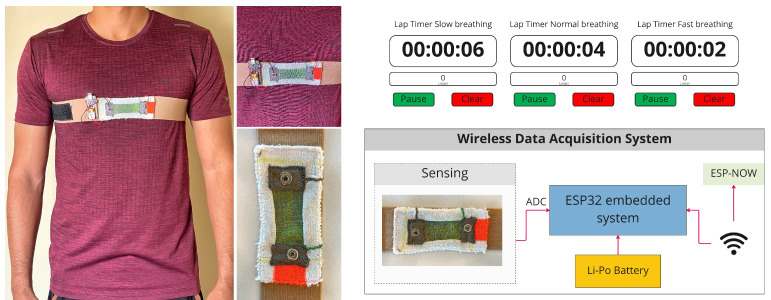
Human testing and data acquisition system.

**Figure 17 biosensors-13-00034-f017:**
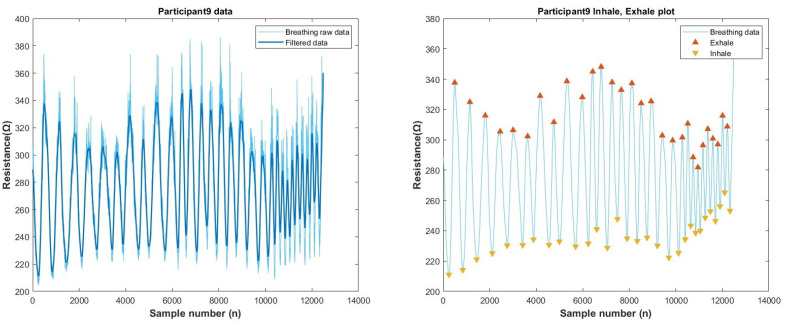
Respiration data analysis (Participant-9). Left-raw and filtered data, right-peak detection for inhale and exhale.

**Figure 18 biosensors-13-00034-f018:**
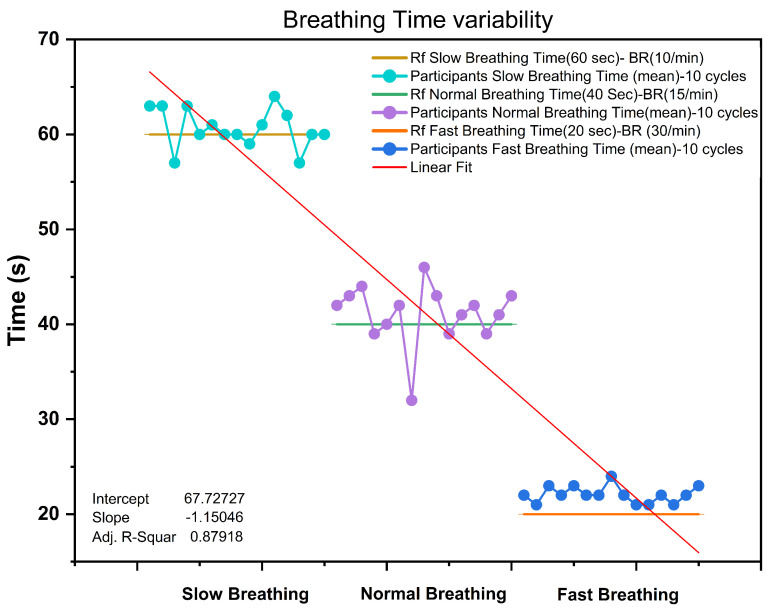
Breathing−time variability of 15 participants during the breathing test (slow, normal, and fast). Rf refers to the referenced breathing time—Slow (60 s), Normal (40 s), Fast (20 s).

**Figure 19 biosensors-13-00034-f019:**
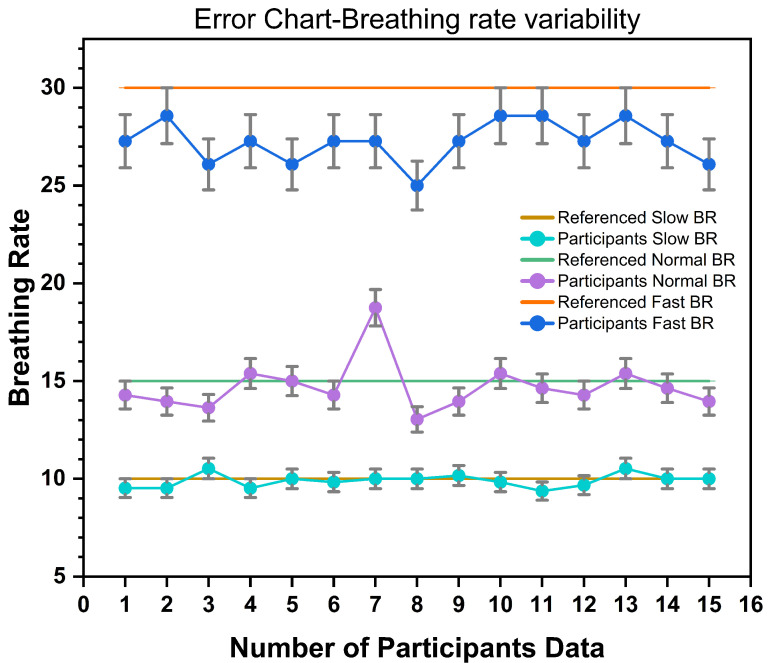
Breathing−rate (BR) variability and error analysis for 15 participants.

**Table 1 biosensors-13-00034-t001:** Different knitted sensor structures and design properties.

Sensor Number	Knitting Type	Description	Dimensions
1	1 × 1 Stripe	Alternate loops of polyester and conductive yarn	1" × 2"
2	Solid	Successive conductive loops	2" × 2"
3	Solid	Successive conductive loops	1.5" × 2"
4	Solid	Successive conductive loops	1" × 2"
5	1 × 2 stripe	alternate loops of two polyester and one conductive yarn	1" × 2"
6	Solid	Hybrid loops of polyester and conductive yarn	1" × 2"

**Table 2 biosensors-13-00034-t002:** Resistance changes during different stretching %.

	Resistance Change (Ω)			
Sensor	10%	15%	20%	25%
1	55.57	79.92	64.66	70
2	34.49	63.16	76.68	81.76
3	22.67	27.88	53.17	59.65
4	18.86	24.10	39.58	42.15
5	29.83	75.08	119.85	79
6	36.42	66.67	81.18	85.54

**Table 3 biosensors-13-00034-t003:** Sensitivity coefficient GF of six sensors.

Sensor Number	Before Durability Cycle Test	After Durability Cycle Test
GF Sensor 1	3.097	2.328
GF Sensor 2	2.611	2.359
GF Sensor 3	3.179	2.776
GF Sensor 4	2.769	2.396
GF Sensor 5	2.694	2.486
GF Sensor 6	3.576	2.502

**Table 4 biosensors-13-00034-t004:** R-squared analysis for different breathing incidents.

Breathing Incident	R-Squared Value
Fast–normal	0.7281
Normal–fast	0.6747
Normal–slow	0.7205
Slow–normal	0.7005
Fast–slow	0.7588
Slow–fast	0.7334

## Data Availability

For further correspondences and data availability please contact to mrumon@uri.edu and kunalm@uri.edu.

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
