# Peer review of "Textile Knitted Stretch Sensors for Wearable Health Monitoring: Design and Performance Evaluation"

_biosensors, 2022, doi:10.3390/bios13010034_

Round 1
Reviewer 1 Report
The authors investigate the stretching behavior of conductive yarns or of polyester yarns covered with a metallic layer and present measurements on respiration experiments to demonstrate the sensor utility as biosensor. All yarns have a knitted structure, in total 6 knitting configurations were studied. The paper includes static and dynamic stretch tests in air before but also after spraying the sensor with water and sodium solution emulating sweating. The paper is quite empirical in nature but rich in experimental results which will be helpful for other researchers in the field. The following points need clarifications:
1- In hysteresis tests (fig. 14 and fig. A11) the resistance at 0% stretch is shown to be higher than the resistance measured under stretch (10-25%). In fig. 9 it appears also that with stretching the resistance is lower from the unstretched case. This is in disagreement with equation (4) where ΔR is positive, fig. 7 and also of the known dependence of a metal's resistance with strain (resistance is increased with strain due to geometrical effect). Please clarify.
2- Some discussion about the physical mechanism that explains the change of resistance in metals should be presented preferably at the point of the text where equations 2-4 are introduced. If the sensors do not follow the known behavior please explain.
3- The gauge factor for the sensors 1-6 should be calculated and reported before durability and after. If changes of GF are observed between these experiments, please discuss.
4- It is understood that the details describing the figures are given in the text. Figure captions are however too short and should be extended to describe better the figures to help the reader.
5- Define the term baseline in 4.1.2 to avoid confusion of the reader.
6- In fig. 8 for each of the sensors (1-6) the curves do not start from the same value of resistance for stretch at 0%. How is that explained?
7- Minor points: In line 512 there is a question mark instead of a figure number. Figures 10 and 8 include too small characters at the horizontal axes.
Author Response
Please see the pdf file.

Reviewer 2 Report
Kindly see the attached files.

Author Response
Please see the pdf file.

Round 2
Reviewer 1 Report
In my previous comment 1 I have witten the following:
In hysteresis tests (fig. 14 and fig. A11) the resistance at 0% stretch is shown to be higher than the resistance measured under stretch (10-25%). In fig. 9 it appears also that with stretching the resistance is lower from the unstretched case. This is in disagreement with equation (4) where ΔR is positive, fig. 7 and also of the known dependence of a metal's resistance with strain (resistance is increased with strain due to geometrical ef ect). Please clarify.
The authors have changed equation (4) so that results are in agreement with fig. 14 and A11 but not in agreeement with results in fig. 7 that shows an increasing resistance with stress. Please clarify.
Reviewer 2 Report
The authors revised the manuscript accordingly, and it can be accepted in its present form.
